# A Universal Method for Modeling and Characterizing Non-Circular Packing Systems Based on *n*-Point Correlation Functions

**DOI:** 10.3390/ma15175991

**Published:** 2022-08-30

**Authors:** Shaobo Sun, Huisu Chen, Jianjun Lin

**Affiliations:** 1Jiangsu Key Laboratory of Construction Materials, School of Materials Science and Engineering, Southeast University, Nanjing 211189, China; 2Key Laboratory of Green Construction and Intelligent Maintenance for Civil Engineering of Hebei Province, School of Civil Engineering & Mechanics, Yanshan University, Qinhuangdao 066004, China

**Keywords:** non-circular particle, *n*-point correlation function, dynamic packing, inter-particle overlapping, microstructure characterization

## Abstract

A universal method for modeling and characterizing non-circular particles is developed. The *n*-point correlation functions (*n* = 1, 2 and 3) are efficiently computed with a GPU parallel computing procedure. An algorithm for dynamic packing of impenetrable non-circular particles is developed based on the fast estimation of overlap information using a one-point correlation function. The packing algorithm is independent of particle shape and proved to be reliable by examples of polygons and super-ellipses. In addition, penetrable packings are generated in an efficient and precise way. Using a two-point correlation function, these non-circular packs are accurately characterized and compared in terms of features such as penetrable and impenetrable, packing fraction and particle shape. In addition, three-point correlation functions are also illustrated and discussed.

## 1. Introduction

Particulate materials are the most widely used heterogeneous materials in the engineering practice [1]. Typical two-phase composites can be seen as particulate or skeletal material [2], which can be geometrically considered as impenetrable particle model and penetrable particle model. Take cementitious materials, for example. Cement paste, mortar or aggregate can be simulated as two-phase particulate material at different scales [3,4,5,6]. The particulate structure can be either a solid phase [7] (impenetrable model) or a porous phase [8] (penetrable model) accordingly. 

### 1.1. Packing of Particulate Model

Impenetrable particle packing is generally called “packing problem”, the final particulate structure of which contains no overlap between particles. The target particulate structure in this paper is that the particles are randomly and discretely distributed, surrounded by the other phase. Therefore, the particulate phase itself is not mechanically stable. For this kind of packing problem, the algorithm of random sequential addition [9] (RSA) is the most commonly used method of random static packing based on Monte Carlo sampling mechanism. RSA refers to a process where particles are randomly and sequentially introduced into a system without overlap to the previous ones. It is a fundamental method for particle packing. However, in the final stage, the elapsed time is unpredictable, and it also has a relatively low “saturated” packing fraction, especially for mono-size particles. i.e., the packing fraction of a saturated random packing for spheres was found to be ϕ=0.382 [10] and for disks ϕ=0.547. To obtain a higher packing fraction, an optimized packing approach is often used in modeling the particulate system [11,12,13,14,15,16,17,18]; most of these works are based on mathematical modeling of the relations between geometric objects and thus reduce the optimal packing problem to a nonlinear programming problem. However, the objective of this approach is to obtain the densest packing system, regardless of considering the real physical movements of particles. There are varieties of packing algorithms that include particle movement and also get a relatively high packing structure, for example, the Discrete Element Method (DEM) [19], the Molecular Dynamics method (MD) [20], the Relaxation Iteration algorithm (RI) [7,21], etc. Compared with other methods, the DEM is able to simulate the real motion of the particles with forces and torques and provides an explicit means of the time-dependent process of the microstructure evolution process. 

Penetrable packing models are easy to generate because neither overlap detection nor particle motion is involved. It can be generated similar to the RSA method without overlapping detection. Based on probability theory, the particle number is pre-calculated according to the targeted packing fraction. However, the resultant packing fraction always fluctuates due to the limited size of system. Percolation thresholds are studied by repeatedly generating the penetrable particles as the porous phase [5,22,23,24], however, the real porosity was not calculated and controlled due to computation cost.

### 1.2. Overlap Determination for Non-Circular Particles

Particle shape is one of the most important factors that determine the macroscopic properties. In recent years, researchers have extended their attention to non-spherical particles, including 2D and 3D, i.e., ellipses/ellipsoids, super-ellipses/super-quadrics, polygons/polyhedrons, composite particles, discretized particles, etc. The DEM originally developed by Cundall and Strack [19] has been used worldwide to study many different shaped particles. In this method, the overlap between particles is considered to represent the particle deformations, which is used to estimate the elastic, plastic and frictional contact forces between particles. Contact detection and overlap determination for disks/spheres are simple and efficient. However, for non-spherical particles, both the contact detection and the overlap determination are complicated, highly dependent on the specific shape characteristic. 

As reviewed in [25,26], composite particles and discretized particles are all able to simulate any non-spherical particles, or any shape theoretically. However, the resolution of these two methods is dependent on the number of units. In practice, it is computationally expensive if we use these two methods to build a well approximated shape. A combination of super-ellipses/super-quadrics and polygons/polyhedrons represents most of the common non-circular/non-spherical particles. Contact detection between particles for these models was well solved by the geometric potential method [27] and the separation axis theorem [28], respectively. Once two particles are confirmed to contact, the following step is to determine the overlap information, including the overlap magnitude, the action point and the normal direction. However, none of these features can be readily defined and obtained by a unified approach due to lack of sound contact theories. Many existing contact models specify the features independently, as we summarized in Table 1.

### 1.3. Characterization of Packs

The particulate system is typically described by various statistical descriptors. Over the past several decades, various descriptors have been proposed to characterize the structure. Among them, the *n*-point correlation functions are able to encompass all the details of the particulate structure [43]. The most common correlation function to characterize particulate structure is the one-point correlation function, also known as the packing fraction in a physical perspective, or the area fraction in 2D and the volume fraction in 3D. Although it is the simplest correlation function, many well-known bounds and prediction models contain only this sole information: for example, the Hashin–Shtirkman (H-S) bounds [44,45], the Mori–Tanaka model [46], the Maxwell–Wagner model [47] and so on. Higher-order bounds for effective permittivity with higher-order correlation functions were derived by Beran [48] and simplified by Torquato [49] and Miltion [50], respectively. Compared with the Hashin–Shtirkman bounds, these higher order bounds added a microstructural parameter ζ. Beran and Molneux [51], McCoy [52] and Milton and Phan-Thien [53] derived higher order bounds for effective shear and bulk moduli with an additional microstructural parameter η. Both ζ and η are triple integrals involving one-, two- and three-point correlation functions, which means that these higher-order bounds contain far more microstructural information than lower-order bounds such as H-S bounds, and enable one to characterize the microstructure more accurately [54]. 

### 1.4. A Universal Method for Modeling and Characterizing

In this paper, one-, two- and three-point correlation functions are computed accurately with the GPU-based parallel method. The one-point correlation function is the probability of finding a given phase at any location. For a particulate system, it can be calculated by judging all points’ location. This process is highly parallelizable, so the GPU-based parallel method will increase the computing speed dramatically. With the advantage of a highly efficient calculation of the one-point correlation function, a novel method is introduced to estimate the comprehensive overlap information between two particles in the dynamic packing of impenetrable particles. This method is described in detail in non-circular models, polygons and super-ellipses. The polygons are composed of vertices and have a unique characteristic of shape corners, while the super-ellipse model is described by a quadric curve, ${\left|x/a\right|}^{2m}+{\left|y/b\right|}^{2m}=1$, where m is the shape parameter characterizing the geometries, a and b are semiaxes. In addition, a penetrable particle model with a precise packing fraction can be obtained with high efficiency. The two-point correlation functions are mainly used to characterize the generated packs because the function curves can be directly compared. The three-point correlation functions are also illustrated in characterization.

## 2. Method Description

### 2.1. Impenetrable Packing Model

#### 2.1.1. Governing Equations

As a well-established dynamic packing method, the DEM has been successfully applied in modeling non-circular/non-spherical particle [25,26,55]. The DEM provides us with an effective way to simulate and analyze the movement of a particulate system; thus, the basic principle of the DEM is utilized in this paper. In the DEM, any particle can have two types of motion: translational and rotational, which are determined by Newton’s second law of motion as given below:(1)$$m_{i}\frac{dv_{i}}{dt}=\sum_{j=1}^{k_{c}}\left(F_{n,ij}+F_{t,ij}\right),$$
(2)$$I_{i}\frac{d\omega_{i}}{dt}=\sum_{j=1}^{k_{c}}\left(M_{n,ij}+M_{t,ij}\right).$$
where $m_{i}$, $I_{i}$, $v_{i}$ and $\omega_{i}$ are the mass, inertia, translational and angular velocities of particle i, respectively, and $k_{c}$ is the number of particles in collision with the particle. As our targeted particulate structure is that the particles are all randomly and discretely distributed, thus in the dynamic packing process, gravity is not considered. The forces considered in this paper are normal force $F_{n,ij}$ and tangential force $F_{t,ij}$. The torques acting on particle i by particle j are consisted of two parts: $M_{n,ij}$, generated by the normal force, and $M_{t,ij}$, generated by the tangential force that caused particle i to rotate. In addition, the normal force $F_{n,ij}$ are also consisted of two parts: the normal elastic force $F_{cn,ij}$ and the viscous damping force $F_{dn,ij}$, and the $F_{dn,ij}$ is used to dissipates the energy of the system.
(3)$$F_{n,ij}=F_{cn,ij}+F_{dn,ij},$$ Similarly, the tangential force $F_{t,ij}$ consists of the tangential elastic force $F_{ct,ij}$ and the tangential damping force $F_{dt,ij}$,
(4)$$F_{t,ij}=F_{ct,ij}+F_{dt,ij}.$$

According to Coulomb’s law, $F_{t,ij}=\min\left\{\left(F_{ct,ij}+F_{dt,ij}\right),\mu \left|F_{n,ij}\right|\right\}$, where μ is the sliding friction coefficient. The equations used to calculate the particle-particle interaction forces and torques are listed in Table 2.

For symbols in this table, k and γ are the spring coefficient and viscous damping coefficient for the contact model, respectively, and the subscripts of n or t indicate the variables are in the normal or tangential directions. They are assumed to be constants during the collisions of particles.

As summarized above in Table 1, for different models, methods of determining overlap information varied considerably. The orientation discretization database solution [42] almost provides a universal method for the different shaped particles, but the databases take time to build and not suitable for many different shapes. Another approach to describe a particle of arbitrary shape is digitization [56], which will take a huge computational effort if accurate results are ensured. We propose a new way of determining overlap information based on the review above. The key features of overlap information (i.e., magnitude, action point and normal direction) are listed in Table 3, and the corresponding contact models are shown in Figure 1.

Based on the features, we determine the normal elastic force. For disks or spheres, the overlap magnitude is defined as a distance $\delta_{n,ij}=R_{i}+R_{j}-R_{ij}$. But for non-circular/non-spherical particles, it is far more effective when using area or volume (in 3D) [55,57]. Hence, we define the overlap magnitude as a function of the overlap area $S_{o}$, which is $\delta_{n,ij}=\sqrt{2S_{o}/\pi }$. Thus, the normal elastic force $F_{cn,ij}$ is defined as:(5)$$F_{cn,ij}=-k_{n,ij}\sqrt{2S_{o}/\pi }n_{ij}.$$

According to the real motion of the particle, it is more reasonable to define the normal damping force $F_{dn,ij}$ on the basis of the relative velocity $v_{ij}$ in the action point $p_{c}$, which is calculated according to Figure 2a.
(6)$$v_{ij}=v_{P_{c},j}-v_{P_{c},i}=v_{j}-v_{i}+\omega_{j}R_{c,ji}-\omega_{i}R_{c,ij},$$
therefore, we obtain the normal damping force,
(7)$$F_{dn,ij}=-\gamma_{n}\left(v_{ij}\cdot n_{ij}\right)\cdot n_{ij}.$$

In addition, the tangential elastic force $F_{ct,ij}$ is defined on the basis of the relative velocity $v_{ij}$ and the time step Δt, and the displacement in the tangential direction is $v_{t,ij}\cdot \Delta t$ in a single time step. According to the resolution of the relative velocity in Figure 2b,
(8)$$F_{ct,ij}=-k_{t,ij}\delta_{t,ij}t_{ij}=-k_{t,ij}\left[v_{ij}-\left(v_{ij}\cdot n_{ij}\right)\cdot n_{ij}\right]\Delta t,$$
Similarly, the tangential damping force can be defined in the opposite direction,
(9)$$F_{dt,ij}=-\gamma_{t}\left(v_{ij}\cdot t_{ij}\right)\cdot t_{ij}=-\gamma_{t}\left[v_{ij}-\left(v_{ij}\cdot n_{ij}\right)\cdot n_{ij}\right].$$

#### 2.1.2. Determination of Overlap Area

As the one-point correlation function is computed with the GPU parallel method, the overlap area can be estimated with high efficiency. Once two particles are detected to overlap, we set a rectangle box to cover them completely, and uniformly generate sampling points, as can be seen from Figure 3. Then all the sampling points are classified by judging their locations. The portion of the points that land in both particles, as shown in Figure 3b, are employed to estimate the overlap information. In this work, the number of the sampling points is not less than $10^{6}$ to guarantee the precision. Then we execute the kernels on the GPU, when a kernel is launched, a grid of threads that are organized in a 3D hierarchy is generated, with each gird being organized into array of thread blocks, and each block containing up to 1024 threads. Therefore, all the sampling points are allocated and judged in millions of threads (i.e., the number of computing cores $N_{thds}=1024\times 1024$ in the case of 2D block and 2D grid). By contrast, each computer has a limited number of CPU cores. To illustrate the parallel speedup ratio of the GPU to the CPU, we tested the speed of the one-point correlation function for the packing of a regular pentagon. An ordinary computer was used, i.e., the CPU is AMD RYZEN 7 3800X, and “GPU” is NVIDIA GeForce RTX 2070. The results are listed in Table 4.

What is worth mentioning is that all points on the “layer” are also classified into the overlap region, to guarantee that, as long as two particles collide theoretically, the overlap will be detected. According to the proportion of points in the overlap region, the overlap area $S_{o}$ is obtained. 

For the non-circular/non-spherical model, a reasonable choice of action point should be the centroid. Under normal circumstances, working out the centroid of the overlapping area/volume is computationally expensive. However, in our method, the overlap region consisted of a set of points, the average location of which is exactly the centroid. Therefore, it can be immediately determined once the overlap points are identified. 

#### 2.1.3. Dynamic Packing Scheme

The flowchart of the dynamic packing scheme on non-spherical particles in this work is shown in Figure 4a.

We set the initial material parameters including: the particle number, the function parameters, the mass, moment of inertia, the particle size and direction angles, etc. In order to eliminate the wall effect, periodic boundary conditions are applied in this paper. First, we generate the initial packings without the overlapping constraints, as shown in Figure 4b. Then direct contact detection between the two particles can be conducted by the geometric potential method and the separation axis theorem for the super-ellipses and the polygons, respectively. For the determined overlapped particles, we estimate the overlap area (volume in 3D) and centroid with the GPU-based parallel method. Forces and torques can be obtained by the equations list above. The positions are calculated by the equations of motion, which can be integrated by the Verlet scheme [58] as follows:(10)$$v_{i}\left(t+\frac{\Delta t}{2}\right)=v_{i}\left(t-\frac{\Delta t}{2}\right)+\frac{F_{i}\left(t\right)}{m_{i}}\Delta t,$$
(11)$$x_{i}\left(t+\Delta t\right)=x_{i}\left(t\right)+v_{i}\left(t+\frac{\Delta t}{2}\right)\Delta t,$$
(12)$$\omega_{i}\left(t+\frac{\Delta t}{2}\right)=\omega_{i}\left(t-\frac{\Delta t}{2}\right)+\frac{M_{i}\left(t\right)}{I_{i}}\Delta t,$$
(13)$$\theta_{i}\left(t+\Delta t\right)=\theta_{i}\left(t\right)+\omega_{i}\left(t+\frac{\Delta t}{2}\right)\Delta t.$$

When the particle information is updated, it goes back to the procedure of contact detection, until no overlap is detected. Finally, we store all the positions and direction angles. The results are visualized as Figure 4c.

### 2.2. Penetrable Packing Model

By using the Poisson limit theorem in statistics, the total inclusion rate for a penetrable system has the quantitative relationship between the inclusion rate and the inclusion number [8]. In this paper, these inclusions can be expressed by particles; thus, the packing fraction ϕ and the particle number $N_{p}$ have the following relationships when $N_{p}=\infty$.
(14)$$\phi =1-e^{-N_{p}}.$$

Therefore, the penetrable models for the various packing fraction can be generated. However, as the particle number is limited, the real packing fraction is only an approximate value, fluctuating around theoretical one. For the continuum percolation models, the pore phase can be simplified to penetrable particles as the porous phase, and by repeatedly generating penetrable models, the percolation threshold of different non-spherical models can be derived. Considering the huge computational cost, the accuracy of the real packing fraction was not considered. In this work, the penetrable packing models are generated efficiently with an accurate packing fraction.

The best approach to acquire the real packing fraction is systematic point sampling, which is equivalent to the one-point correlation function. In our approach, due to the high efficiency of the one-point correlation function, one can repeatedly generate penetrable packings and at the same time calculate the real packing faction, until the one within close tolerance appears. The flowchart of the packing algorithm is shown in Figure 5a.

For example, a penetrable super-ellipse and a regular square with a packing fraction of 0.8 are generated with different precision, the time elapsed in increasing as upgrading precision, as shown in Table 5.

## 3. *n*-Point Correlation Functions

The *n*-point correlation function is a generalized definition of the correlation functions. The higher-order correlation functions are able to provide more information about the geometric features of a generation, for example, the standard two-point correlation function, the three-point correlation function, and various other two-point correlation functions such as Lineal-Path Function, Chord-Length Density Function, Pore-Size Functions and so on. In this work, only the standard one-, two- and three-point correlation functions, which are included in higher-order parameters ζ and η, are concerned.

The *n*-point correlation function is the probability that the *n* given points with the locations $x_{1},x_{2},\ldots ,x_{n}$ will be in the same phase i,
(15)$$S_{n}^{\left(i\right)}\left(x_{1},x_{2},\ldots ,x_{n}\right)=P\left\{I^{\left(i\right)}\left(x_{1}\right)=1,I^{\left(i\right)}\left(x_{2}\right)=1,\ldots ,I^{\left(i\right)}\left(x_{n}\right)=1\;\right\},$$
and it can be expressed as the expectation (or average) of the multiplication of the indicator functions at the *n* locations,
(16)$$S_{n}^{\left(i\right)}\left(x_{1},x_{2},\ldots ,x_{n}\right)=\langle I^{\left(i\right)}\left(x_{1}\right)I^{\left(i\right)}\left(x_{2}\right)\cdots I^{\left(i\right)}\left(x_{n}\right)\rangle ,$$
where the angular bracket 〈⋯〉 denotes the expectation or the ensemble average, $I^{\left(i\right)}$ represents indicator function for phase i,
(17)$$I^{\left(i\right)}\left(x\right)=\{\begin{array}{c} 1,\;\;\;\;\;x\in V_{i},\\ 0,\;\;\;\;\;x\in \bar{V_{i}}, \end{array}$$
where $V_{i}$ is the region occupied by phase i with the packing fraction $\phi_{i}$. One can define the one-, two- and three-point correlation functions when *n* = 1, 2 and 3:(18)$$S_{1}^{\left(i\right)}\left(x\right)=\langle I^{\left(i\right)}\left(x\right)\rangle =P\left\{I^{\left(i\right)}\left(x\right)=1\;\right\},$$
(19)$$S_{2}^{\left(i\right)}\left(x_{1},x_{2}\right)=\langle I^{\left(i\right)}\left(x_{1},x_{2}\right)\rangle =P\left\{I^{\left(i\right)}\left(x_{1}\right)=1,I^{\left(i\right)}\left(x_{2}\right)=1\;\right\}\;,$$
(20)$$S_{3}^{\left(i\right)}\left(x_{1},x_{2},x_{3}\right)=\langle I^{\left(i\right)}\left(x_{1},x_{2},x_{3}\right)\rangle =P\left\{I^{\left(i\right)}\left(x_{1}\right)=1,I^{\left(i\right)}\left(x_{2}\right)=1,I^{\left(i\right)}\left(x_{3}\right)=1\;\right\}.$$

For example, for a two-phase composite material, the gray phase represents the “aggregate”, while the white phase is the matrix. Schematic representation of the one-, two- and three-point correlation functions are shown in Figure 6a. $S_{aa}$ is another form of $S_{2}^{\left(i\right)}\left(r\right)$ when i represents the aggregate phase. In addition, $S_{mm}$ means that phase i of interest is the matrix, but there exists another function $S_{am}$, which means that one point lands on the aggregate, and the other point lands in the matrix. Therefore, it is obvious that: (21)$$S_{aa}+S_{mm}+S_{am}=1\;.$$

Similarly, $S_{aaa}$ means $S_{3}^{\left(i\right)}\left(r_{1},r_{2},\theta \right)$ when the phase i of interest is the aggregate. In this work, particles in the realizations are randomly distributed and the periodic boundary conditions are considered. It is reasonable to assume all generations are statistically homogeneous and isotropic; therefore, these functions can be obtained by the random sampling technique. These correlation functions can be calculated as follows.

The one-point correlation function $S_{a}$ can be calculated by systematic point sampling method, and the points can be generated as both randomly distributed or uniformly arrayed. Considering the overlap area calculation that was mentioned above and was shown in Figure 3b, the uniformly arrayed points are used as the sampling points. Moreover, the number of points in each sampling is $10^{6}–10^{8}$ accordingly.

The two-point correlation function $S_{aa}\left(r\right)$ is the probability of the two points at the phase a. The result of $S_{aa}\left(r\right)$ is not a single number, because there is a probability value for each length r. Theoretically r=[0,∞], for the purpose of comparison, only the selected values of r are calculated. For example, if one wants to compute ζ and η, the r value often started at an extremely small value [54]. In our work, the start point of the segment is uniformly arrayed, and the end point is determined according to r and a random angle. The number of sampling lines $N_{line}$ is chosen $10^{5}–10^{7}$ accordingly. When r=0 and r=∞, the limiting values are obtained:(22)$$S_{2}^{\left(i\right)}\left(0\right)=\phi_{i},$$
(23)$$\underset{r\to \infty }{\lim}S_{2}^{\left(i\right)}\left(r\right)=\phi_{i}^{2}.$$

The three-point correlation function $S_{aaa}\left(r_{1},r_{2},\theta \right)$ is the probability of the three points at the aggregate phase. The sampling triangle can be determined by $r_{1}$, $r_{2}$ and θ. Theoretically $r_{1},r_{2}=\left[0,\infty \right]$, θ=[0,π]. $S_{aaa}\left(r_{1},r_{2},\theta \right)$ is not commonly used because it is computationally expensive. For example, when $r_{1},r_{2}$ and θ are all divided into 100 values, there will be $100^{3}$ different patterns of the sampling triangle; for each pattern, the number of the sampling number $N_{3pt}$ in this work is $10^{5}–10^{7}$ accordingly. In addition, the limiting values are obtained when $r_{1},r_{2}\to 0$ and $r_{1},r_{2}\to \infty ,$ θ≠0:(24)$$\underset{r_{1},r_{2}\to 0}{\lim}S_{3}^{\left(i\right)}\left(r_{1},r_{2},\theta \right)=\phi_{i},$$
(25)$$\underset{r_{1},r_{2}\to \infty ,\theta \neq 0}{\lim}S_{3}^{\left(i\right)}\left(r_{1},r_{2},\theta \right)=\phi_{i}^{3}.$$

## 4. Discussion

Based on the packing algorithm, the various non-circular packing systems are generated. The *n*-point correlation functions describe the probabilities of the different phase encounters and other geometric features and aim to encompass all details of the packing system, i.e., the two-point correlation function has been extensively used in the characterization of short-range information. In what follows, we characterize the packing systems with the different packing fraction and the particle shape using the two-point correlation functions. In addition, the three-point correlation function is illustrated and discussed.

### 4.1. Packing Fraction

We generated the realizations with the packing fraction $\phi_{a}=0.1$ to $\phi_{a}=0.8$ using the packing method described above in this work. Figure 7a illustrates examples of the generations for the super-ellipse (a/b, m=2) with the packing fraction $\phi_{a}=0.1,\;0.4\;\mathrm{and}\;0.8$. Figure 7b correspondingly presents the penetrable ones.

The two-point correlation function $S_{aa}\left(r\right)$ of the particulate phase for the super-ellipse with the packing fraction $\phi_{a}=0.1\sim 0.8$ is shown in Figure 7c,d. When r=0, only two types of the function exist: $S_{aa}$ and $S_{mm}$, at this time $S_{aa}+S_{mm}=1$ which is actually a one-point correlation function. As r grows, $S_{aa}\;$is decreased due to the emergence of $S_{am}$. By comparing Figure 7c,d, we can clearly see that for impenetrable models, the decay of $S_{aa}$ is faster, and obvious oscillations begin to show up when $r\geq D_{eq}$. At that point, the decay corresponds to their long-ranger values, while the penetrable results did not show clear oscillations. It is worth noting that when the packing fraction is very low, i.e., $\phi_{a}=0.1$, $S_{aa}$ results are basically the same for the impenetrable and the penetrable packing models, because the generations are similar, as shown in Figure 7a. In addition, the speed at which the fluctuations level off is different for $S_{aa}$ in the impenetrable system. 

### 4.2. Particle Shape

To study the effect of the particle shape of $S_{aa}\left(r\right)$, we generated a series of packing models with the same packing fraction $\phi_{a}=0.5$. For simplicity without loss of generality, we used the special cases of the super-ellipse. First, we let m=1, super-ellipse is equivalent to ellipse, we compared $S_{aa}\left(r\right)$ for different aspect ratio κ. Then we let κ=1, and computed $S_{aa}\left(r\right)$ for the different shape parameter m. The results are shown in Figure 8a, a growing κ led to a faster decay, and the fluctuations were smaller. There will be an obvious difference when κ is changing from 1 to 5, while when κ is fixed, as we see in Figure 8b, the overall pattern of $S_{aa}\left(r\right)$ is similar and the slope before first trough of wave is almost identical, but a visible difference appears near the first peak of wave.

Then we compared $S_{aa}\left(r\right)$ of the different regular polygons: triangle, cube and hexagon. As we can observe in Figure 8c, $S_{aa}\left(r\right)$ of the regular cube has the largest trough, because in this paper, the equivalent circular diameter $D_{eq}$ is used for non-circular particles, and the regular square has the longest line segment of these three shapes. Moreover, the regular square has the fastest decay. We also computed $S_{aa}\left(r\right)$ in penetrable models, and an example is shown in Figure 8d. A relatively smaller difference can be observed compared with the impenetrable ones.

Roundness, also called circularity, is a 2D measure of how closely the shape of an object approaches that of a mathematically perfect circle, which is defined as the perimeter ratio of the particle and the circle with the same area [59]. The roundness of the super-ellipse could be any value only by changing the aspect ratio κ, as shown in Table 6. 

Therefore, it is reasonable to compare the two particles with the same roundness, i.e., the regular square and the ellipse with an aspect ratio κ=2.283670. $S_{aa}\left(r\right)$ are calculated for both impenetrable and penetrable models, and the resulting comparisons are shown in Figure 9a,b. We can see a clear difference in the impenetrable model and a visible distinction in the penetrable model, which means the two-point correlation functions are able to describe more morphological features besides roundness.

### 4.3. Three-Point Correlation Function

The three-point correlation functions $S_{aaa}\left(r_{1},r_{2},\theta \right)$ for the generations of regular squares with the packing fraction $\phi_{a}=0.7$ are presented in Figure 10.

We only show some selected values of θ for illustration. It can be observed that the one- and the two-point correlation function are contained within the three-point correlation function, and the function can be degenerated. When $r_{1}=r_{2}=0$, $S_{aaa}\left(r_{1},r_{2},\theta \right)$ degenerates to the one-point correlation function $S_{a}$. We let $\theta =0,\;r_{1}=r_{2}\neq 0$, as shown in Figure 10. The zig zag diagonal line is a special case of two-point correlation function, and it has the steepest slop. In this function, the two end points of $r_{1}$ and $r_{2}$ coincide and have to be judged twice. According to Equation (21), it is obvious that for the different phase of interest, $S_{aaa}\left(r_{1},r_{2},\theta \right)$ in two-phase materials,
(26)$$S_{aaa}+S_{aam}+S_{amm}+S_{mmm}=1,$$
and in this situation, when it degenerates to $S_{aa}\left(r\right)$,
(27)$$S_{aa}+S_{am}+S_{am}+S_{mm}=1,$$
the $S_{am}$ is doubled, which explains the fast decay of $S_{aa}$ and the zig zag line afterward. When $r_{1}=0$ or $r_{2}=0$, it degenerates to the standard two-point correlation function $S_{aa}\left(r\right)$. It can be observed that the two-point correlation function $S_{aa}$ is only a small section of the three-point correlation function $S_{aaa}\left(r_{1},r_{2},\theta \right)$, even for the one selected θ.

When calculating the function values for $S_{3}^{\left(i\right)}\left(r_{1},r_{2},\theta \right)$, $N_{3pt}$ random sampling triangles are used. Each sampling triangle has $N_{r_{1}}\times N_{r_{2}}\times N_{\theta }$ shapes. The accuracies are often verified by their degeneracies because the fundamentals of computing the one-, two- and three-point correlation functions are the same, that is to determine a single point’s location. By comparing $S_{aaa}\left(r\right)$ to $S_{aa}\left(r\right)$, as shown in Figure 8c, the three-point correlation characterizes the structure more comprehensively, while the two-point correlation can describe the obvious features, i.e., particle shape. 

## 5. Conclusions

The *n*-point correlation functions, including the one-, two- and three-point correlations in this paper, are utilized for modeling and characterizing a non-spherical packing system.

A dynamic packing algorithm for impenetrable non-circular particles was developed based on the DEM. In this algorithm, a novel method of determining overlap information (i.e., overlap area and centroid) for non-circular particles was developed by means of efficiently calculating the one-point correlation function. In addition, a penetrable non-spherical packing algorithm with high precision and efficiency was proposed via the one-point correlation function, and it is applicable to arbitrary shape in principle. 

With the packing algorithms, packs of non-circular particles, both impenetrable and penetrable, with a different packing fraction (*ϕ* = 0.1 − 0.8) and various geometry shapes (regular polygons and super-ellipses), have been generated. The two-point correlation function are chosen as a statistical descriptor in this work for characterizing the packs. For the impenetrable models, clear differences can be observed with different packing fractions or various geometry shapes, even if the roundness is the same. The differences in corresponding penetrable models are less pronounced. The three-point correlation function is illustrated in three selected *θ*. It characterizes far more details than two-point correlation function. Moreover, with the efficient computation of the one-, two- and three-point correlations, an effective material behavior of the particulate systems can be predicted by third-order bounds, which will provide us a straightforward route to quantitative characterization.

## Figures and Tables

**Figure 1 materials-15-05991-f001:**
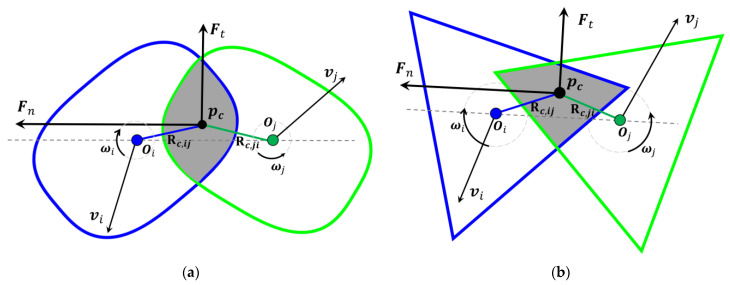
Contact model of non-circular particles: (**a**) Super-ellipse *a*/*b* = 1.5, *m* = 1.5; (**b**) Regular triangle.

**Figure 2 materials-15-05991-f002:**
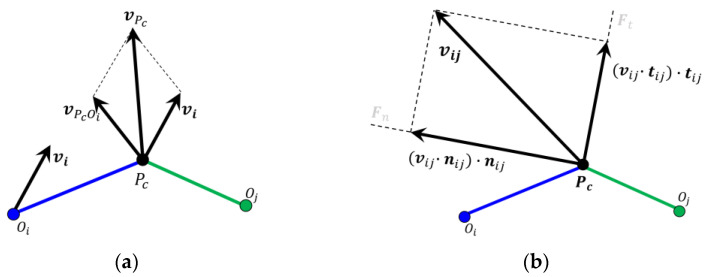
(**a**) Relative velocity $v_{ij}$ in the action point *p_c_*; (**b**) Resolution of the relative velocity.

**Figure 3 materials-15-05991-f003:**
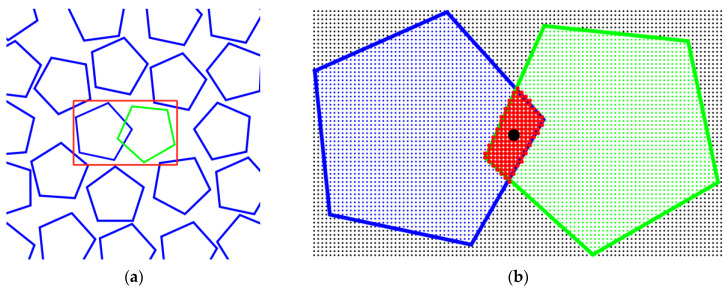
(**a**) A rectangle box surrounding two overlapping particles; (**b**) Point sampling details based on one-point correlation function: the red dots indicate the overlapped ones, and the black dot is the centroid.

**Figure 4 materials-15-05991-f004:**
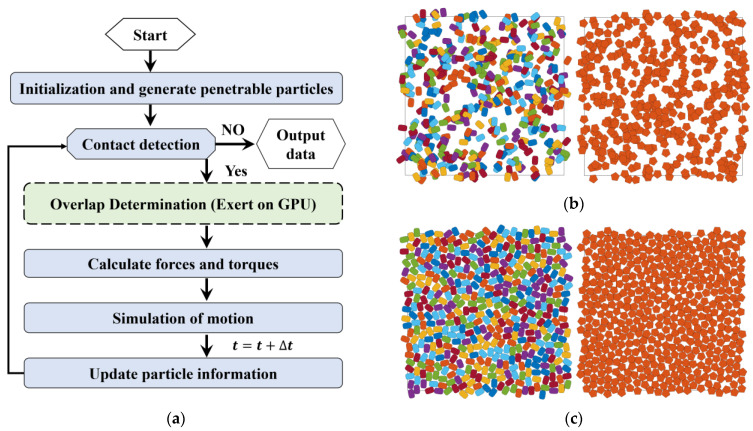
An example of dynamic packing super-ellipses (packing fraction of 80%, container = 100 × 100, *D_eq_* = 5, *a*/*b* = 1.5 and *m* = 1.5) and polygon (regular pentagon). (**a**) Flow chart for the packing process; (**b**) Initial state of packing, highly penetrable, particle number *N_p_* = 407; (**c**) Final state of packing, particle number *N_p_* = 451 and 450 (including periodic ones) for super-ellipse and regular pentagon, respectively.

**Figure 5 materials-15-05991-f005:**
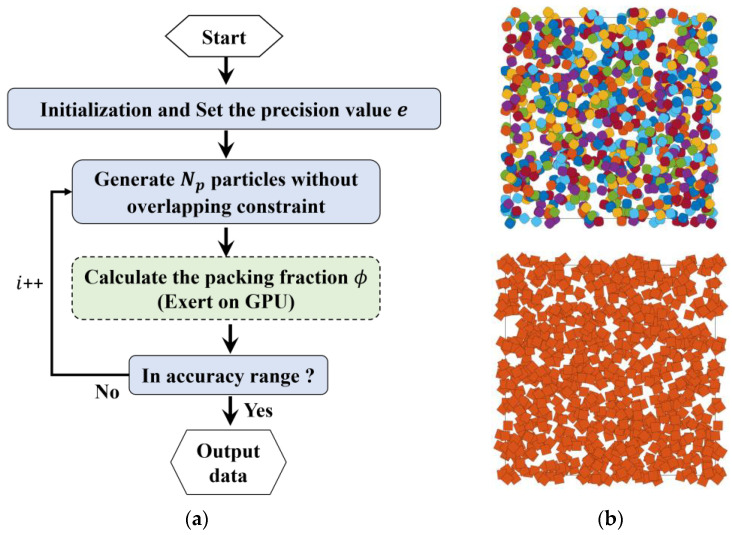
(**a**) Flowchart of a highly efficient packing with accurate packing fraction; (**b**) Examples of penetrable packing models with a packing fraction *ϕ* = 0.8 ± *e*, where the precision *e* = 10^−5^. For both generations, the particle number (*N_p_* = 819) is the same because the equivalent diameter *D_eq_* is used for non-circular particles.

**Figure 6 materials-15-05991-f006:**
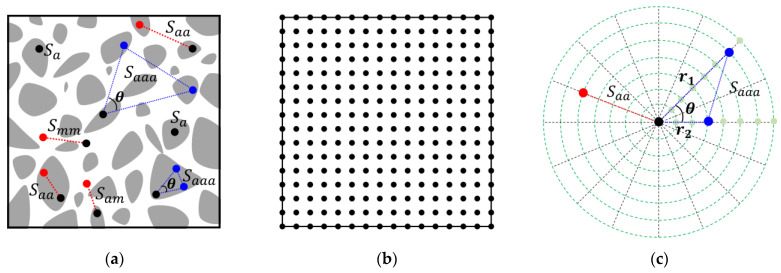
(**a**) Illustration of one-, two- and three-point correlation functions; (**b**) Systematic point sampling: uniformly arrayed sampling points; (**c**) Sampling lines and sampling triangles for two- and three-point correlation function, respectively.

**Figure 7 materials-15-05991-f007:**
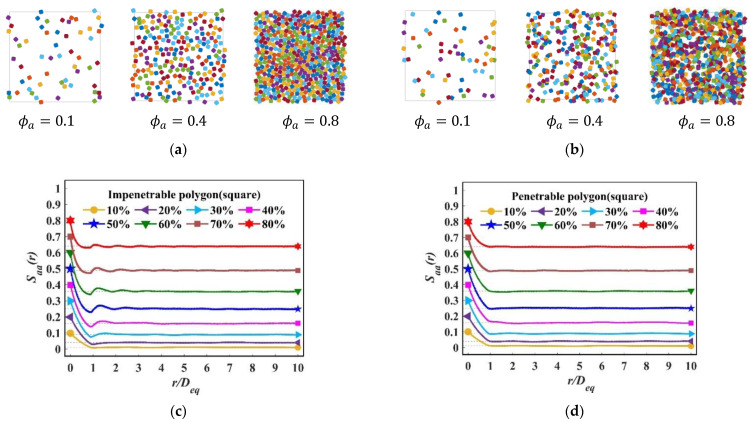
(**a**) Examples of impenetrable packs of super-ellipse, *a*/*b* = 1, *m* = 2; (**b**) Examples of penetrable packings for super-ellipse, *a*/*b* = 1, *m* = 2; (**c**) *S_aa_*(*r*) for impenetrable super-ellipse, with various packing fraction; (**d**) *S_aa_*(*r*) for penetrable super-ellipse, with various packing fraction.

**Figure 8 materials-15-05991-f008:**
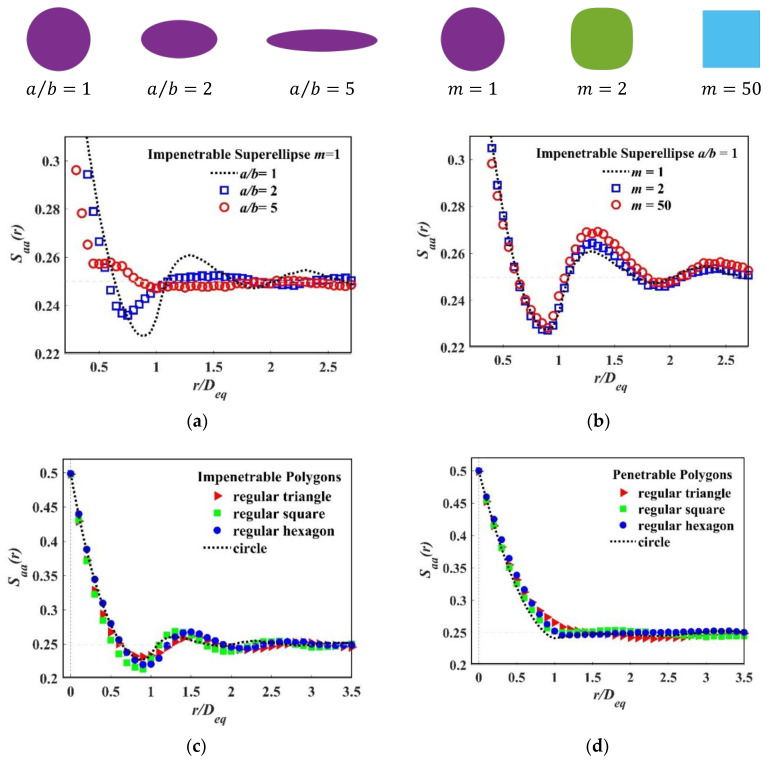
*S_aa_*(*r*) of different particle shapes: (**a**) Selected aspect ratio *κ* from 1 to 5; (**b**) Selected shape parameter *m* from 1 to 50; (**c**) Selected regular polygons, from triangle to circle, which can be treated as a polygon with infinite sides; (**d**) Selected penetrable models: from triangle to circle.

**Figure 9 materials-15-05991-f009:**
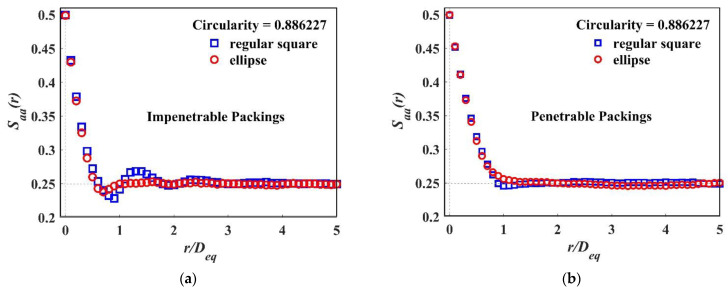
*S_aa_*(*r*) of different particle packing systems with the same circularity and the same packing fraction *ϕ**_a_* = 0.5: (**a**) Impenetrable packing system of regular square and ellipse; (**b**) Penetrable packing system of regular square and ellipse.

**Figure 10 materials-15-05991-f010:**
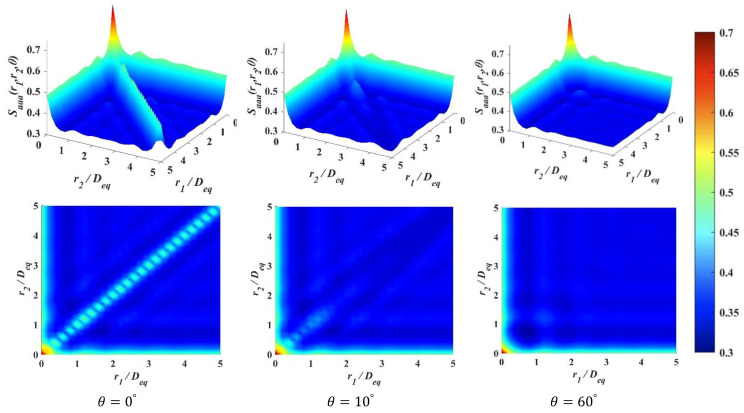
Illustration of *S_aaa_*(*r*_1_, *r*_2_, *θ*) for regular squares with selected *θ* = 0°, 10° and 60°.

**Table 1 materials-15-05991-t001:** Selected methods on overlap determination of non-spherical particles.

Method	Model	Magnitude	Action Point	Normal Direction
Intersection [29,30]	Ellipse	Distance	Midpoint of intersection line	Perpendicular to the intersection line
Geometric potential [31,32,33]	Super-ellipse Super-quadric	Distance	Midpoint of two points with lowest geometric potential	Connecting two points with lowest geometric potential
Common normal [34,35]	Super-quadric	Distance	Midpoint of two surface points sharing a common normal	Connecting two points sharing a common normal
Intersection [36]	Polygon	Distance	Midpoint of intersection line	Perpendicular to the intersection line
Energy-based normal contact model [37]	Polygon	Area	Midpoint of two intersecting points for corner-corner contact	A direction that can decrease the contact energy with maximum rate
Intersection [38,39]	Polyhedron	Volume	Centroid of overlap volume	Perpendicular to the plane that is taken as the least-squares fit of hull intersection curve
Inner potential particles [40,41]	Polygon Polyhedron	Distance	Analytic center of linear inequalities	Gradient vector of an inner potential particle
Orientation discretization database solution [42]	Polygon, Ellipse, Ellipsoid and others.	Area(2D) Volume(3D)	Average of centers of overlap cells	Averaged normal vector of the cell at the surface of the particle

**Table 2 materials-15-05991-t002:** Components of forces and torques acting on particle.

Forces and Torques	Symbols	Equations
Normal elastic force	$F_{cn,ij}$	$-k_{n,ij}\delta_{n,ij}n_{ij}$
Normal damping force	$F_{dn,ij}$	$-\gamma_{n}\left(v_{ij}\cdot n_{ij}\right)\cdot n_{ij}$
Tangential elastic force	$F_{ct,ij}$	$-k_{t,ij}\delta_{t,ij}t_{ij}\;\left(\left|F_{t,ij}\right|\leq \mu \left|F_{n,ij}\right|\right)$
Tangential damping force	$F_{dt,ij}$	$-\gamma_{t}\left(v_{ij}\cdot t_{ij}\right)\cdot t_{ij}\;\left(\left|F_{t,ij}\right|\leq \mu \left|F_{n,ij}\right|\right)$
Coulomb friction force	$f_{t,ij}$	$-\mu \left|F_{n,ij}\right|\cdot t_{ij}\;\left(\left|F_{t,ij}\right|>\mu \left|F_{n,ij}\right|\right)$
Torque by normal force	$M_{n,ij}$	$R_{c,ij}\times \left(F_{cn,ij}+F_{dn,ij}\right)$
Torque by tangential force	$M_{t,ij}$	$R_{c,ij}\times \left(F_{ct,ij}+F_{dt,ij}\right)$

**Table 3 materials-15-05991-t003:** Contact overlap features used in this paper.

Magnitude	Action Point	Normal Direction
**Area** (Volume in 3D): Ratio of sampling points	**Centroid**: Average of sample points	Pass the action point and parallel with the line connecting two centroids of particles

**Table 4 materials-15-05991-t004:** Total consumption times of the CPU and GPU in calculation of one-point correlation function for packing system of regular pentagon.

Number of Points	10^5^	10^6^	10^7^
*t_GPU_* (s)	0.024 s	0.031 s	0.462 s
*t_CPU_* (s)	0.482 s	0.625 s	5.972 s

Note: the packing system is closely related to the execution time. In this test, the packing fraction *ϕ_a_* = 0.5, total particle number *N_p_* = 254.

**Table 5 materials-15-05991-t005:** The relationship between precision and elapsed time.

Precision (−)	10^−2^	10^−3^	10^−4^	10^−5^
*t_superellipse_* (s)	0.13	0.78	6.46	43.25
*t_square_* (s)	0.06	0.21	2.71	18.52

**Table 6 materials-15-05991-t006:** The relationship between precision and elapsed time.

Roundness	0.777560	0.886227	0.929950	0.952313	1
*κ* of Ellipses	3.417562	2.283670	1.883470	1.675994	1
Regular polygons	Triangle	Square	Pentagon	Hexagon	Circle

## Data Availability

Not applicable.

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
