# Peer review of "A Universal Method for Modeling and Characterizing Non-Circular Packing Systems Based on n-Point Correlation Functions"

_materials, 2022, doi:10.3390/ma15175991_

Round 1

Reviewer 1 Report

The manuscript entitled "A universal method for modeling and characterizing non-spherical packing systems based on n-point correlation functions" outlines an effective method for the generation of models of nonpenetrating and penetrating packing for non-spherical particles. The manuscript highlights the role of GPU parallel processing in making this an efficient method. I would like to see more detail on this part of the process and how it is implemented and empirical conformation of the acceleration of this stage compared to a linear processing of the step. 

A range of 1-, 2-, 3-point corrections are utilised to characterise these packings, again I'm not clear on what information is obtained from these calculations. Figure 8 (a) shows a change in curve shape with aspect ratio but little difference with shape parameter. What information does this give the user about the system? Identification of the particle shape? I think the work would be improved if the purpose of these correlations could be clarified. 

Finally there a number of incomplete figure captions (Figures 2,9) and errors (Table 4 is included twice, ref 49 is incomplete) in the text and these will need revision before the manuscript is suitable for publication. 

Reviewer 2 Report

The authors present a novel approach to modeling non-circular particulate structures. An algorithm for dynamic packing of non-circular particles is developed based on the fast estimation of overlap information using a 1-point correlation function. The examples of packing 2D polygons and super-ellipses are provided and analyzed. The paper fits the scope of the journal and present interesting results. English is good. I think the paper can be recommended for publication after considering the following minor comments.

1.      Constructions presented in the paper are justified and implemented for 2D case. Worth reflecting this in the title of the paper. Correspondingly, it is better using the term “non-circular” instead of “non-spherical”.

2.      Along with DEM approach to packing problems which simulates physical movements of particles to get the dense structure, the so-called optimized packing approach is also used in modeling particulate structures. Here the objective is to get the densest packing without overlapping by minimizing a certain objective, e.g., area/volume of the sample container (square or cuboid). Optimized packing ellipsoids are considered, e.g., in Packing ellipsoids in an optimized cylinder, European Journal of Operational Research, 285(2):429- 443, 2020. Packing different non-circular 2D objects were considered, e.g., in The Dotted-Board Model: A new MIP model for nesting irregular shapes," International Journal of Production Economics, 145(2), 478-487, 2013; Packing circular-like objects in a rectangular container, Journal of Computer and Systems Sciences International, 54(2):259-267, 2015. I think one or two paragraphs with a short review of the corresponding approaches would be useful to highlight the place of the paper in the overall packing publication stream.

3.      Line 112. Please revise the definition of the super-ellipse in (x,y) coordinates.

4.      Line 150 “For a clear view, we list in Table 3, the corresponding contact models are shown in Figure 1.”. Not clear, please rephrase.

5.      Reference [49], publication information is missing.

Round 2

Reviewer 1 Report

The authors have revised the manuscript following the reviewers comments. I am happy with their responses to the points raised and feel they have improved the quality of the manuscript in doing so.